# Specialized Pro-Resolving Mediators in Neuroinflammation: Overview of Studies and Perspectives of Clinical Applications

**DOI:** 10.3390/molecules27154836

**Published:** 2022-07-28

**Authors:** Mariarosaria Valente, Marta Dentoni, Fabrizio Bellizzi, Fedra Kuris, Gian Luigi Gigli

**Affiliations:** 1Neurology Unit, Dipartimento di Area Medica (DAME), University of Udine, 33100 Udine, Italy; dentoni.marta@spes.uniud.it (M.D.); bellizzi.fabrizio@spes.uniud.it (F.B.); kuris.fedra@spes.uniud.it (F.K.); 2Clinical Neurology Unit, Department of Neurosciences, S. Maria della Misericordia University Hospital, 33100 Udine, Italy

**Keywords:** specialized pro-resolving mediators, resolvins, maresins, annexins, lipoxins, protectins, glial cells, neuroinflammation, neurodegeneration, cerebrovascular disorders, multiple sclerosis, dementia

## Abstract

Specialized pro-resolving mediators (SPMs) are lipid mediators derived from poly-unsaturated fatty acids (PUFAs) which have been demonstrated to have an important role in the inflammation environment, preventing an overreaction of the organism and promoting the resolution of inflammation. Our purpose was to point out the current evidence for specialized pro-resolving mediators, focusing on their role in neuroinflammation and in major neurological diseases.

## 1. Specialized Pro-Resolving Mediators: Metabolism, Receptors, Pathways

### 1.1. Overview on Specialized Pro-Resolving Mediators

Inflammation is a cascade event preserved along the evolution from the first multicellular precursor organisms to humans. Its main role is to defend tissues from an insulting agent, such as microbes or direct damage, enabling in most cases a natural return to homeostasis. If inflammation is not someway stopped, it can lead to serious consequences, such as uncontrolled edema [1]. 

For many years, it was assumed that inflammation was a self-limiting process [1]. However, recent discoveries have shown the presence of an active de-escalation process, promoted by a class of molecules, namely specialized pro-resolving mediators (SPMs). From the beginning of the inflammation process, SPMs reach the site of edema, either transported by blood flow or produced within the inflammatory tissue [1]. Since chronic and/or uncontrolled inflammation plays a key role in a variety of diseases (such as cardiovascular diseases, metabolic syndrome, and neurological diseases), SPMs have a potential therapeutic role. In particular, SPMs are lipid mediators (LMs) derived from PUFAs (poly-unsaturated fatty acids), such as AA (Arachidonic Acid), EPA (eicosapentaenoic acid), DHA (docosahexaenoic acid) and *n*-3 DPA (*n*-3 docosapentaenoic acid). The properties of ω-3 fish oil fatty acids in human disease and physiology may in part be explained by the formation of autacoids derived from PUFAs [1]. SPMs include lipoxins, resolvins, protectins and maresins, as well as newly identified cysteinyl-conjugated SPMs (cys-SPMs) and *n*-3 DPA-derived SPMs [2]. In the following paragraphs, each group of SPMs will be analyzed.

#### 1.1.1. Lipoxins

Lipoxins (LX) LXA4 and LXB4 [1,3] were the first discovered SPMs. Lipoxins derive from eicosanoids thanks to a mechanism of lipo-oxygenation. Eicosanoids in turn derive from AA, an ω-6 fatty acid implied in inflammation. AA is converted into LXA4 and LXB4 via 5- lipoxygenase and 15- lipoxygenase. Lipoxins are produced by leukocytes in transcellular biosynthesis steps during interactions between leukocytes and mucosal cells or platelets [1]. Initially, Lipoxins were believed to be agents of anti-inflammation, and their pro-resolution role has only recently been discovered. Aspirin can trigger their biosynthesis thanks to its capacity to promote the formation of LMs via lipo-oxygenation [4].

#### 1.1.2. Resolvins

Resolvins are generated in inflammatory exudates during the phase of resolution. They derive from ω-3 fatty acids EPA and DHA, forming E series (RvE) and D series (RvD) resolvins, respectively. Their synthesis is also promoted by aspirin (likewise Lipoxins) [1]. Resolvins act in several ways in order to interrupt the inflammation cascade. In particular, they [1]:inhibit the production of pro-inflammatory mediators, such as chemokines and cytokinesenhance scavenging of pro-inflammatory chemokinespromote the recruitment of monocytes and phagocytes’ clearance via the lymphatic systemlimit PMN (polymorphonuclear cells) migration and infiltrationFocusing on the subclass of Resolvins, we can find [2]:E-series Resolvins: RvE1, RvE2, RvE3 and the recent RvE4;D-series Resolvins: RvD1, 17R-ResolvinD1, RvD2, RvD3 and 17R-Resolvin D3, RvD4, RvD5.

#### 1.1.3. Protectins

Protectins consist of Protectin D1/Neuroprotectin D1 (PD1/NPD1). They are biosynthesized from DHA via the 15-LOX mechanism. It can be found in human cell types, murine exudates and brain tissue; in this last case, it is called “NeuroprotectinD1” (NPD1), whereas PD1 operates in peripheral tissue. PD1/NPD1 has neuroprotective properties in the brain, retina and Central Nervous System (CNS). Its aspirin-triggered epimer, 17R-NPD1, has the same actions as NPD1 in controlling PMN, enhancing macrophage functions and attenuating experimental stroke [2].

#### 1.1.4. Maresins

Maresins were first identified in human macrophages, in a pathway initiated by 12-LOX. Their name derives from an acronym: Macrophage Mediators in Resolving Inflammation. Maresin1 (MaR1) is able to promote the regeneration of tissues in an experimental model of simple organisms (planaria) with a strong capability of regeneration. In human cells, it is produced by platelets and PMN interactions. MaR1 promotes tissue regeneration and repair and has a neuroprotective role [2].

#### 1.1.5. Recently Discovered SPMs

Recently discovered peptide-lipid conjugated SPMs include cysteinyl-SPMs (cys-SPMs). They consist of three series of SPMs, each one with three bioactive members: maresin conjugates in tissue regeneration (MCTR), protectin conjugates in tissue regeneration (PCTR) and resolving conjugates in tissue regeneration (RCTR). They show pro-repair and pro-regenerative actions [2]. For the sake of completeness, it is important to mention n3-DPA-derived SMPs: RvDn-3 DPA, MaRn-3 DPA and PDn-3 DPA, 13-series resolvins (RvTs). They share the potent actions of DPA and EHA-derived SPMs in the resolution of systemic inflammation and neuro-inflammation. RvTs’ biosynthesis is promoted by atorvastatin via S-nitrosylation of cyclooxygenase-2 (COX-2) [2].

### 1.2. Receptors and Pathways

It is important to emphasize that these endogenous mediators of resolution do not act thanks to an “inhibition” of inflammation pathways: instead, they actively promote specific pathways in order to obtain a return to homeostasis. There are specific G-protein-coupled seven-transmembrane receptors (GPCR) activated by SPMs [1]. Every single class of SPM demonstrates stereoselective activation of its own GPCR. SPMs show affinities for ligand-receptors in the nano-picomolar range, thus demonstrating a potent action in vitro and in vivo [2]. 

Resolvin E1 (RvE1) acts via ChemR23 (GPCR for RvE1). It is also a partial agonist on the LTB_4_ (leukotriene B4) receptor (BLT1), activated by LTB_4_ as well. Nevertheless, RvE1 has a different mechanism of action, which is a time and dose-dependent phosphorylation of Akt and p70S6K (ribosomal protein S6 kinase) via ChemR23 [1].

Resolvin D1 (RvD1) binds two separate GPCR on human leukocytes: ALX/FPR2 (LXA_4_ receptor) and GPR32 (GPCR for RvD1) [1]. ALX/FPR2 receptors can also be activated by Annexin-1 and Chemerin [5]. Deficits in ALX/FPR2 in experimental models (mice) amplify cardiomyopathy, age-related obesity, and leukocyte-directed endothelial dysfunction [6].

MaR1 can activate two classes of receptors: leucine-rich repeat-containing G protein-coupled receptor 6 (LGR6), a phagocyte’s receptorretinoic acid-related orphan receptor α (ROR-α), a liver macrophages’ nuclear receptor

Stimulating the LGR6 receptor, MaR1 can promote phagocytosis, efferocytosis, and the phosphorylation of select proteins [7]. NPD1/PD1’s receptor, GPR37, increases intracellular Ca^2+^ in macrophages and promotes phagocytosis [8]. RvD5n-3 DPA binds an orphan receptor, GPR101, with high stereospecificity [2]; in experimental KO models of GPR101, there is a lack of protective action of RvD5n-3 in inflammatory arthritis [9].

These receptors demonstrate overlapping actions (for example, ALX, GPR18, LGR6 and GPR101 can promote calcium mobilization via cAMP signal) and distinct actions too; thus, they could act in tandem to promote defense from injury, inflammation, and infection [2]. 

### 1.3. Mechanism of Action

The first signs of inflammation response are vasodilation and changes in vessel permeability. These factors not only permit the recruitment of cells implied in the inflammatory response but also give substrates for the biosynthesis of important molecules, such as SPMs [1]. Apparently, ω-3 PUFAs, AA, EPA and DHA can be found within inflammatory exudates during very early phases, as demonstrated in various works [10,11]. Therefore, the inflammation response is counterbalanced early by pro-resolution mediators. This avoids an excess of an inflammatory response that can be disruptive for the organism and for the tissue itself [1].

## 2. Specialized Pro-Resolving Mediators and Neuroinflammation

### 2.1. Neuroinflammation and Its Resolution

While inflammation is usually a self-limiting physiological process, when persistent or dysregulated it can become harmful to human tissues; if this happens within the CNS it is referred to as neuroinflammation and many studies proved that chronic neuroinflammation could ultimately lead to neurodegeneration [12,13,14]. In this picture, an emerging concept is the resolution of neuroinflammation which contributes to brain homeostasis; a great deal of attention has been paid to the topic in the last few years. The main actors of this specular process are the so-called SPMs, whose characteristics have been explained in the previous chapter. In the last decade, several research groups started to investigate the role of SPMs in the nervous tissue as regulators of the inflammation process that may contribute to the crosstalk between glial cells and neurons in several neurological pathologies [15]. 

### 2.2. The Role of Glial Cells in Neuroinflammation and the Contribution of SPMs

Nervous tissue is composed of about 100 billion neurons and 80 to 100 billion glial cells, namely ectoderm-derived astrocytes and oligodendrocytes, and mesoderm-derived microglial cells. Astrocytes play a key role in the metabolism and metabolic support of nervous parenchyma and specifically neurons, i.e., lactate shuttle, the glutamate–glutamine cycle, and ketone bodies supply. Neuroinflammation has lately been interpreted as a condition of metabolic imbalance and energetic depletion, both in the acute and chronic settings. It hence derives that glial cells play a crucial role in the control of neuroinflammation, by regulating nervous tissue metabolism. 

As demonstrated, brain tissue contains high levels of PUFAs, mainly DHA and AA, which are the principal precursors of SPMs. The main PUFA source is unesterified plasma fatty acid pool rather than endogenous synthesis; such a source is severely impacted by dietary supply according to studies conducted on rodents [16,17]. Interestingly, the hippocampus and prefrontal cortex contain the highest DHA content while the hypothalamus has the lowest [15]. As for their proportion of representation in the human brain, astrocytes contain 10–12% of DHA, oligodendrocytes 5%, and microglial cells up to 2% [18]. Astrocytes, the most abundant glial cells present in the nervous tissue, take part in many vital processes, such as the migration of developing axons and certain neuroblasts, the regulation of blood flow, electrolyte homeostasis, blood–brain barrier (BBB), and synapse function. Moreover, they seem to be the main glial cells involved in neuroinflammation, although they show significant diversity in this process. For instance, they express high levels of the ALX/FPR2 receptor, which has a central role in the regulation of astrogliosis, an active inflammatory path that leads to neural protection, repair and ultimately to glial scarring [15]. LXA_4_ and RvD1, the two SPMs that bind this receptor, promote the inhibition of astrocytes’ pro-inflammatory activities [19]. Moreover, it has been observed that peripheral RvD1 administration in brain injury models improved its functional recovery through an ALX/FPR2-regulated pathway probably induced by astrocytes [15]. Another important receptor expressed by astrocytes and playing an important role in neuroprotection is ChemR23/ERV1, expressed in the human hippocampus, which binds RvE1: animal studies demonstrated that peripheral administration of RvE1 in Alzheimer’s disease (AD), in combination with LXA_4_, reduced astrocyte activation [20]. Other receptors involved in the neuroprotection and resolution of inflammation are GPR37, GPR18 and LgR6, whose expression in astrocytes is challenged, and further studies both in vivo and in vitro are needed on this subset. Besides their main function of myelin synthesis, oligodendrocytes, the second most represented cell population in the CNS, may play a role in the resolution of neuroinflammation thanks to the latest evidence on their active production of immune-regulatory factors or their receptors [21]. Comparing oligodendrocytes with astrocytes, ALX/FPR2 is not expressed by these cells; the only SPMs receptor identified seems to be GPR37 [22]. On the other hand, microglia, the immune cells of the CNS, thanks to their very physiological role, seem to express all the known SPM receptors and are susceptible to the effects of different SPMs categories (lipoxins, RvE, RvD, protectins and maresins) [15,23]. Nonetheless, the cellular origin of SPMs in these cells, as in astrocytes and oligodendrocytes, has not yet been demonstrated and only a few in vitro studies have tried to investigate it [16]. 

## 3. Specialized Pro-Resolving Mediators and Potential Applications in Neuroinflammatory Conditions

### 3.1. Specialized Pro-Resolving Mediators in Ischemic Stroke and Cerebrovascular Events

The concept of ischemic stroke has been expanded to include not only what happens inside the vessel, but also in the surrounding environment, the so-called “neurovascular unit”, which includes the interaction between glia, neurons, vascular cells, and matrix components; after the acute event, secondary neuroinflammation takes place, bringing about detrimental effects producing further injury and neuronal death, and promotion of recovery [24]. Several studies have investigated the possible role of pro-resolving mediators in improving post-stroke prognosis; however, they have mostly been conducted on rodents, and applications in humans remain speculative and in need of further research. Table 1 provides a summary of in vivo studies on SPMs in ischemic stroke and cerebrovascular events.

#### 3.1.1. Resolvins in Ischemic Stroke and Cerebrovascular Events

Though preliminary studies indicate a decrease in the risk of cardiovascular diseases thanks to *n*-3 PUFAs supplementation, large double-blind studies did not show clear beneficial effects; however, PUFAs may play a role both through the reduction of pro-inflammatory factors, as well as through the stimulation of the resolution of inflammation [25]. The human body metabolizes *n*-3 PUFAs into RvD2 via the lipoxygenase pathway. Exogenous supply of RvD2 via intraperitoneal injection in a middle cerebral artery occlusion (MCAO) mouse model was able to reverse induced brain injury, including infarction, inflammatory response, brain edema, and neurological dysfunction [26]. Apparently, the capacity of *n*-3 PUFAs to generate RvD2 was reduced by middle cerebral artery occlusion/reperfusion (MCAO/R), making their supplementation less effective than direct RvD2 injection: in the early brain ischemia/reperfusion (I/R) injury process, the metabolic processing of fish oil (especially DHA) may be blocked. In the same animal model, neutrophil membrane-derived nanovesicles loaded with RvD2 were shown to alleviate inflammation and protect the mouse brain from ischemic stroke injury, thus providing a possible therapeutic strategy [27]. Despite what has just been said about PUFAs, acute post-ischemic administration of triglyceride emulsions containing only DHA (tri-DHA) conferred neuroprotection against hypoxic-ischemic injury in neonatal mice [28,29]. RvD1 may play a role in modulating stroke risk factors by preventing atherosclerosis: supplementation with exogenous RvD1 improved plaque stability in fat-fed Ldlr-/- via an increase in fibrous cap thickness and decreased lesional oxidative stress and necrosis [30]. Further studies demonstrate how RvD2 and RvE1 supply may significantly decrease or slow down atherosclerotic changes as well [25]. Post-stroke blood levels of RvD1 have also been found to correlate with cognitive performance: in a prospective study assessing the impact of eicosanoids on cognitive function in stroke survivors, prostaglandin E2, 9S-, 13S-HODE and RvD1 were all strongly associated with the post-stroke cognitive impairment, while RvD1 only correlated with better cognitive performance [31]. 

#### 3.1.2. Maresins in Ischemic Stroke and Cerebrovascular Events

The intracerebroventricular injection of MaR1 may play a protective role against I/R injury by inhibiting pro-inflammatory reactions and NF-kB p65 activation and nuclear translocation: in a MCAO mice model, MaR1 significantly reduced the infarct volume and neurological defects, protecting the brain tissue and neurons from injury [32]. MaR1 treatment also attenuated cerebral I/R injury by reducing inflammatory responses and mitochondrial damage via the activation of SIRT1 signaling [33].

#### 3.1.3. Annexins in Ischemic Stroke and Cerebrovascular Events

The pro-resolving protein Annexin A1 (AnxA1) has been studied too; targeting the AnxA1/Fpr2/ALX pathway may represent another novel treatment strategy for resolving thrombo-inflammation [34,35]. Macrophages can differentiate into two subtypes, depending on cytokines and chemokines production during inflammation response. In particular, different chemokines can attract Th1 and Th2 or T regulatory (Tr) cells, and this response is integrated by M1 and M2 macrophages in circuits of amplification and regulation of T-cell responses. M1 macrophages are effector cells that kill microorganisms and tumor cells, producing a multitude of proinflammatory cytokines. On the other hand, M2 cells tune inflammatory responses and adaptive Th1 immunity, scavenge debris, and promote angiogenesis, tissue remodeling and repair [36]. Ac2-26 (annexin/lipocortin 1-mimetic peptide) administered to a transient MCAO/R mouse model was shown to modulate microglial/macrophage polarization towards M2 anti-inflammatory phenotype and alleviate subsequent cerebral inflammation; this was achieved by regulating the FPR2/ALX-dependent AMPK-mTOR pathway [37]. The same study pointed to plasma AnxA1 as a potential biomarker for the outcomes of acute ischemic stroke patients receiving endovascular thrombectomy. AnxA1 administration was shown to be beneficial in intracerebral hemorrhage, attenuating neuroinflammation via the AnxA1/FPR2/p38 signaling pathway [38]. In addition, in cerebral I/R injury AnxA1 may shift the platelet phenotype from pro-pathogenic to regulatory: it was able to reduce the propensity of platelets to aggregate and cause thrombosis by affecting integrin (αIIbβ3) activation [39].

#### 3.1.4. Lipoxins in Ischemic Stroke and Cerebrovascular Events

LXA4 is another potent anti-inflammatory mediator exerting a neuroprotective effect following a cerebrovascular event. It has been postulated to regulate microglial M1/M2 polarization after cerebral I/R injury via the Notch signaling pathway, and to downregulate the expression of the proinflammatory cytokines IL-1β and TNF-α [40]. The intracerebroventricular injection of LXA4 or its synthetic analogs was shown to decrease infarct volumes and improve neurological function in mice. One study pointed to Nrf2 upregulation being involved in the neuroprotective effects of LXA4; such effects were partially blocked by Boc2, a specific antagonist of the LXA4 receptor (ALXR) [41]. However, LXA4 induced Nrf2 expression and its nuclear translocation, as well as HO-1 expression and GSH synthesis; the latter two effects were not blocked by Boc2, indicating that Nrf2 upregulation may be ALXR independent. In addition, the PPAR agonist rosiglitazone has been shown to be neuroprotective by increasing LXA4 and reducing leukotriene B4 (LTB4) in experimental stroke [42]. 

As LXA4 is rapidly inactivated, potent analogs have been synthesized, including BML-111. Post-ischemic treatment with BML-111 significantly reduced infarct size, decreased vasogenic edema, protected against BBB disruption, and reduced hemorrhagic transformation in rats [43]. Similarly, post-ischemic, intravenous treatment with BML-111 for 1 week was shown to induce early protective effects, reducing infarct volume, and improving sensorimotor function at 1 week; however, it did not reduce infarct size or improve behavioral deficits 4 weeks after ischemic stroke [44]. Another stable synthetic analog of LXA4 is lipoxin A4 methyl ester (LXA4 ME). Intracerebroventricular injection in I/R injury mice ameliorated neurological dysfunctions, reduced infarction volume, attenuated neuronal apoptosis and had overall an anti-inflammatory effect [45,46]. It was also shown to reduce BBB dysfunction and MMP-9 expression while increasing TIMP-1 expression [47]. One study suggested that intracerebroventricular injection for two consecutive weeks in mice after the acute event could alleviate spatial learning and memory impairments, thus exerting beneficial effects on the cognitive impairment induced by chronic cerebral hypoperfusion. This was likely achieved through attenuating oxidative injury and reducing neuronal apoptosis in the hippocampus with the activation of the ERK/Nrf2 signaling pathway [48]. Interestingly, one of the few studies on LXA4 conducted in humans showed that the levels of LXA4 in the acute phase of ischemic stroke were significantly reduced in post-stroke cognitive impairment patients compared with those with no cognitive impairment upon Mini-Mental State Examination (MMSE) testing [49]. LXA4 has also been reported to reduce neuroinflammation by activating FPR2 and inhibiting p38 in a rat model of subarachnoid hemorrhage (SAH); intracerebroventricular injection of exogenous LXA4 reduced brain water content and BBB leakage, and improved neurological function, memory and learning after the event [50]. An amelioration of endothelial dysfunction, microflow recovery, and suppression of neutrophil infiltration was also shown, possibly involving the LXA4/FPR2/ERK1/2 pathway [51].

#### 3.1.5. Protectins in Ischemic Stroke and Cerebrovascular Events

Protectins, also called neuroprotectins, may play a role in stroke too. NPD1 was demonstrated to reduce infarct volume, inhibit the activation of NF-kB, and reduce the expression of COX-2 and infiltration polymorphonuclear leukocytes [52]. The intracerebroventricular injection of NPD1 in the rat I/R injury model has been shown to significantly reduce infarct volume and improve neurological scores, through the inhibition of calpain-mediated TRPC6 proteolysis and the subsequent activation of CREB via the Ras/MEK/ERK pathway [53]. Infarct size reduction in aged rats via the activation of the Akt and p70S6K pathways has also been demonstrated [54]. NPD1 seems to counter uncompensated oxidative stress by upregulating ring finger protein 146 (Iduna) in neurons and astrocytes, which facilitates DNA repair and protects against cell death; in fact, Iduna is usually downregulated in the penumbra after cerebral ischemia [55]. NPD1 may also work on mitochondria-related cell death pathways, which play a major role in ischemic brain injury. Following NPD1 acute intraperitoneal injection in mice, ischemic core expansion was prevented by about 40%; brain mitochondria showed a preserved membrane structure, together with a reduction of mitochondrial BAX translocation and activation [56]. NPD1 administration has also been demonstrated to promote neurogenesis and angiogenesis, BBB integrity, penumbra protection and subsequent long-term neurobehavioral recovery after experimental ischemic stroke [57]. By administering aspirin plus DHA, the synthesis of aspirin-triggered NPD1 (AT-NPD1) in the brain was discovered; the total chemical synthesis of this molecule and usage in mice MCAO model was shown to promote sustained neurobehavioral recovery, reduce infarct volume and brain edema, and protect white matter [58].molecules-27-04836-t001_Table 1Table 1Summary of in vivo studies on SPMs in ischemic stroke and cerebrovascular events.ReferenceType of StudyAnimal ModelPro-Resolving MediatorDelivery (Or Measurement If the Study Was Non-Interventional)OutcomeZuo et al., 2018 [26]Animal studyMCAO mouse modelRvD2intraperitoneal ↓ infarction, inflammation, edema, and neurological dysfunction; compared with ω-3 fatty acid oral supplements, better rescue effect on cerebral infarctionDong et al., 2019 [27]Animal studyMCAO mouse modelRvD2Intravenous infusion of RvD2-loaded nanovesicles ↓ inflammation;↑ neurological functionFredman et al., 2016 [30]Animal studyfat-fed Ldlr-/- miceRvD1Immunoprecipitation injection↓ atherosclerosisKotlęga et al., 2021 [31]Human study-RvD1blood levels of endogenous pro-resolving mediatorsPost-stroke blood levels of RvD1 correlated with a better cognitive performanceXian et al., 2016 [32]Animal studyMCAO mouse modelMaR1Intracerebroventricular ↓infarct volume and neurological defects by inhibiting NF-kB p65 functionXian et al., 2019 [33]Animal studyMCAO mouse modelMaR1Intracerebroventricular ↓ inflammation and mitochondrial damage via activation of SIRT1 signalingVital et al., 2020 [34]Animal studyLipopolysaccharide and sickle transgenic mice models of thrombo-inflammationAnxA1 mimetic peptide Ac2-26Intravenous ↓ thrombo-inflammation via Fpr2/ALX receptor and ↓ platelet aggregationGavins et al., 2007 [35]Animal studyMCAO in wild-type or AnxA1−/− miceAnxA1 mimetic peptide Ac2-26Intravenous ↓ inflammation via receptors of the FPR familyXu et al., 2021 [37]Animal studyMCAO mouse modelAnxA1 mimetic peptide Ac2-26Intravenous ↓ inflammation by regulating the FPR2/ALX-dependent AMPK-mTOR pathwayDing et al., 2020 [38]Animal studyCollagenase-induced ICH mouse modelRecombinant human AnXA1Intracerebroventricular↓ inflammation via the FPR2/p38/COX-2 pathwaySenchenkova et al., 2019 [39]Animal studyMCAO in wild-type or AnxA1−/− miceWhole protein AnXA1Intravenous↓ platelet aggregation by affecting integrin (αIIbβ3) activationLi et al., 2021 [40]Animal studyMCAO mouse modelLXA4Intracerebroventricular↓ proinflammatory cytokines and regulate microglial M1/M2 polarization via the Notch signaling pathwayWu et al., 2013 [41]Animal studyMCAO mouse modelLXA4Intracerebroventricular↓infarct volume and ↑ neurological function through Nrf2 upregulationHawkins et al., 2014 [43]Animal studyMCAO mouse modelLXA4 analog BML-111Intravenous ↓ infarct size, edema, BBB disruption, and hemorrhagic transformationHawkins et al., 2017 [44]Animal studyMCAO mouse modelLXA4 analog BML-111Intravenous ↓ infarct volume; and↑ neurological function at 1 week.No reduction of infarct size or improvement of behavioral deficits 4 weeks after ischemic strokeWu et al., 2010 [45]Animal studyMCAO mouse modelLXA4 MEIntracerebroventricular ↓ proinflammatory cytokines, neurological dysfunctions, infarction volume, and neuronal apoptosisYe et al., 2010 [46]Animal studyMCAO mouse modelLXA4 MEIntracerebroventricular↓ proinflammatory cytokines, neurological dysfunctions, infarction volume, and neuronal apoptosisWu et al., 2012 [47]Animal studyMCAO mouse modelLXA4 MEIntracerebroventricular↓ BBB dysfunction and MMP-9 expression; ↑ TIMP-1 expressionJin et al., 2014 [48]Animal studyBCCAOLXA4 MEIntracerebroventricularAmelioration of cognitive impairment via ↓oxidative injury and ↓neuronal apoptosis in the hippocampus with the activation of the ERK/Nrf2 signaling pathwayWang et al., 2021 [49]Human study-LXA4, RvD1, RvD2, RvE1, MaR1blood levels of endogenous pro-resolving mediators↓ LXA4 in patients with post-stroke cognitive impairmentGuo et al., 2016 [50]Animal studyendovascular perforation model of SAHExogenous LXA4Intracerebroventricular↓ neuroinflammation by activating FPR2 and inhibiting p38Liu et al., 2019 [51]Animal studyendovascular perforation model of SAHRecombinant LXA4Intracerebroventricular↓ endothelial dysfunction and neutrophil infiltration, possibly involving the LXA4/FPR2/ERK1/2 pathwayYao et al., 2013 [53]Animal studyMCAO mouse modelNPD1Intracerebroventricular ↓ infarct volume and ↑ neurological scores through inhibition of calpain-mediated TRPC6 proteolysis and activation of CREB via the Ras/MEK/ERK pathwayEady et al., 2012 [54]Animal studyMCAO mouse modelNPD1Intravenous ↓ infarct size in aged rats via activation of Akt and p70S6K pathwaysBelayev et al., 2017 [55]Animal studyMCAO mouse modelDHA (NPD1 precursor)Intravenous ↓ oxidative stress by upregulating ring finger protein 146 (Iduna) in neurons and astrocyteZirpoli et al., 2021 [56]Animal studyUnilateral cerebral hypoxia-ischemia injury mouse modelNPD1Intraperitoneal↓ ischemic core expansion, preserved mitochondrial structure and ↓ BAX translocation and activationBelayev et al., 2018 [57]Animal studyMCAO mouse modelNPD1Intracerebroventricular↑ neurogenesis and angiogenesis, BBB integrity, and long-term neurobehavioral recoveryBazan et al., 2012 [58]Animal studyMCAO mouse modelAT-NPD1Intravenous ↓ infarct volume and brain edema; ↑ neurobehavioral recoveryAnXA1: Annexin A1; AT-NPD1: aspirin-triggered NPD1; BBB: Blood–Brain Barrier; BCCAO: bilateral common carotid artery occlusion; DHA: docosahexaenoic acid; FPR: formyl-peptide receptor; LXA4: Lipoxin A4; LXA4 ME: Lipoxin A4 Methyl Ester; MaR1: Maresin1; MCAO: middle cerebral artery occlusion; NPD1: Neuroprotectin D1; RvD1: Resolvin D1; RvD2: Resolvin D2; RvE1: Resolvin E1; SAH: Sub Arachnoid Hemorrhage.

### 3.2. Specialized Pro-Resolving Mediators in Neurological Immune-Mediated Disorders

Multiple sclerosis (MS) is a neuroinflammatory disease in which unresolved and uncontrolled inflammation leads to a pathological disease state, thus representing a classical model of chronic inflammation; in this context, SPMs could be instrumental in resolving the pathologic inflammation. However, there are minimal data available on the functional status of SPMs in MS; it seems that SPMs have neuroprotective action in MS by exerting pro-resolving effects in the pre-clinical model; however, little is known about the direct effect of SPMs on oligodendrocytic or neuronal cells [59]. Table 2 provides a summary of in vivo studies on SPMs in neurological immune-mediated disorders.

#### 3.2.1. Annexins in Neurological Immune-Mediated Disorders

The potential role of SPMs has been studied in experimental autoimmune encephalomyelitis (EAE), a mouse model of MS. Across the literature, the role of AnxA1 appears to be contentious. However, it is worth underlying that AnxA1 likely exerts a dual function on the innate and adaptive immune systems; in the innate immune system, endogenous AnxA1 plays an anti-inflammatory role that controls events occurring early in the inflammatory process; on the contrary, in the adaptive immune system its role is controversial, and it may as well be pro-inflammatory [60,61]. Dated works demonstrating annexin-1 immunoreactivity in plaque lesions in both experimental mice and MS patients led to hypothesizing a possible contribution to anti-inflammatory processes [62,63,64]. Another study pointed to the potential therapeutic benefit of annexin-1 administration in MS, as intracerebroventricular administration in EAE rats significantly reduced neurological severity, and immunoneutralization of endogenous brain annexin-1 failed to exacerbate the clinical features of EAE [65]. In another work, the potential modulatory role of AnxA1 in the development of EAE was investigated, and a direct correlation between AnxA1 T cells expression and severity of disease was shown. MOG35-55-induced EAE development was impaired in AnxA1 null mice, which showed decreased signs of the disease compared to wild type mice at the peak and reduced infiltration of T cells in the spinal cord [60]. Moreover, reduced in vitro recall proliferative response to MOG35-55 in Annexin A1 null T cells was demonstrated, with a significantly reduced Th1/Th17 phenotype, as compared to wild type cells [60]. Authors thus concluded that the identification and generation of neutralizing antibodies against AnxA1 could play a therapeutic role in MS. A more recent study tried to shed some light on the ability of AnxA1 to influence T cell effector function in relapsing/remitting MS (RRMS); by measuring circulating expression levels of AnxA1 in RRMS patients, it was found that they are inversely correlated with disease score and progression [61]. In addition, at the cellular level, there was impaired AnxA1 production in CD4+CD252 conventional T and CD4+RORgt+ T (Th17) cells from RRMS subjects that were associated with an increased migratory capacity in an in vitro BBB model. Authors associated AnxA1 anti-inflammatory action with the STAT3 signaling pathway.

#### 3.2.2. Resolvins in Neurological Immune-Mediated Disorders

As far as concerns the therapeutic potential of RvD1 in EAE mice, oral administration was very effective in attenuating disease progression by suppressing autoreactive T cells and inducing an M2 phenotype of monocytes/macrophages and resident brain microglial cells, though not affecting the number of infiltrating cells [66]. 

#### 3.2.3. Lipoxins in Neurological Immune-Mediated Disorders

LXA4 may play a role in MS as well. Intraperitoneal injection of LXA4 was shown to ameliorate EAE clinical symptoms and inhibit CD4+ and CD8+ T cell infiltration into the CNS; in addition, LXA4 potently reduced encephalitogenic Th1 and Th17 effector functions, both in vivo and in isolated human T cells from healthy donors and patients with RRMS [67]. The same study demonstrated that LXA4 affects the spinal cord lipidome by significantly reducing the levels of pro-inflammatory LMs during EAE. 

#### 3.2.4. Maresins in Neurological Immune-Mediated Disorders

A recent paper analyzed SPMs in active brain lesions, serum, and peripheral blood mononuclear cells (PBMCs) in MS patients and in the spinal cord of EAE mice, showing that levels of MaR1 and other SPMs were below the limit of detection or not increased in mice [68]. Similarly, they were undetected in serum and active brain lesion samples of MS patients, which may be linked to impaired expression of the enzymes involved in the biosynthetic pathways of SPMs. When exogenous MaR1 was administered to mice, various pro-inflammatory cytokines were suppressed, the number of Th1 cells was reduced and the number of Tregs increased, while macrophages underwent polarization towards an anti-inflammatory phenotype [68]. 

#### 3.2.5. Differential Expression of SPMs in MS

The possibility of non-exhaustive or possibly ‘delayed’ resolution pathways in MS was also suggested by a study observing that LM pathways are regulated differentially in the cerebrospinal fluid (CSF) of MS patients, depending on disease severity [69]. Specifically, in patients with highly active MS, RvD1 was significantly upregulated and NPD1 was detected in this group only. In line with these results, another study group showed distinct LM profiles that significantly correlated with disease severity in MS patients’ peripheral blood [70]. In particular, relapsing and progressive MS patients were associated with high eicosanoid levels, whereas the majority of pro-resolving LM were either significantly reduced or below limits of detection and correlated with disease progression. Furthermore, the expression of several enzymes and associated receptors involved in SPM biosynthesis was found to be reduced in the blood-derived leukocytes of MS patients. These findings support the idea that differentially expressed mediators, such as LXA4, LXB4, RvD1 and NPD1 reduced MS-derived monocyte activation and cytokine production and inhibited inflammation-induced BBB dysfunction and monocyte trans-endothelial migration. The same study group recently presented data at the 35th Annual Congress of the European Committee for Treatment and Research in Multiple Sclerosis (ECTRIMS, 2019) suggesting impaired production of SPMs in MS patients. Comprehensive metabolipidomics profiling was used to identify the spectrum of LM signatures in the CSF of patients across different clinical courses, including relapsing, remitting, and progressive modes of MS; CSF analysis revealed lower levels of LXB4 and RvD3 in different clinical courses of the disease (unpublished) [59].

#### 3.2.6. SPMs in Demyelinating Disorders Other Than MS

Other than the previously discussed findings in MS, few studies on neurological demyelinating disorders and pro-resolving mediators have been published. A recent paper pointed to inflammation resolution impairment in neuromyelitis optica spectrum disorders (NMOSD), showing that RvD1 levels were significantly decreased, whereas leukotrienes B4 (LTB4) levels were significantly increased in the CSF of NMOSD patients [71]. Furthermore, AQP4-IgG titer was negatively correlated with RvD1 levels in the CSF of NMOSD patients, indicating such antibodies may contribute to increased and unresolved inflammation. SPMs have been investigated in experimental autoimmune neuritis (EAN) too, a model of acute inflammatory demyelinating polyradiculoneuropathy. Annexin-1 expression was found to be increased in the inflamed sciatic, which may indicate immunoregulatory functions in-situ and contribute to the termination of the autoimmune response [72]. In addition, in the EAN model RvD1, its synthetic enzyme and receptor were found to be increased in the peripheral nervous system (PNS) during the recovery stage of EAN; intraperitoneal RvD1 injection led to macrophage phagocytosis of apoptotic T cells in PNS, thereby upregulating TGFβ by macrophages, increasing local Treg cell counts, and finally promoting inflammation resolution and disease recovery [73]. molecules-27-04836-t002_Table 2Table 2Summary of in vivo studies on SPMs in neurological immune-mediated disorders.ReferenceType of StudyModelPro-Resolving MediatorDelivery (Or Measurement If the Study Was Non-Interventional)OutcomePaschalidis N et al., 2009 [60]Animal studyMOG34-55 -induced EAE in AnxA1 null mice compared to MOG34-55 -induced EAE in control miceAbsence of AnxA1 expressionMeasurement of disease activity in spinal cord; lymph-node cells (respectively, by isolation of T-cells and/or fixation with haematoxylin and eosin; and by test ELISA for Th1/Th17 cytokine profile)↓ signs of the disease in AnxA1 null mice compared to wild type mice ↓ infiltration of T cells in the spinal cord of AnxA1 null mice compared to wild typeHuitinga I et al., 1998 [65]Animal studyEAE rats (MS mouse model)AnxA1Intracerebroventricular administration↓ neurological severityPoisson LM, 2015 [66]Animal studyEAE rats (MS mouse model)RvD1Oral administrationAttenuation of disease progression by suppressing autoreactive T cells and inducing an M2 phenotype of monocytes/macrophages and resident brain microglial cellsDerada Troletti C et al., 2021 [67]Animal studyEAE rats (MS mouse model)LXA4Intraperitoneal injectionImprovement of EAE clinical symptoms and inhibit CD4+ and CD8+ T cell infiltration into the CNSDerada Troletti C et al., 2021 [67]In vivo and in vitro studyHuman T cells from healthy donors and patients with relapsing-remitting MSLXA4Measurement of T-cell functions↓ encephalitogenic Th1 and Th17 effector functionsSánchez-Fernández A et al., 2022 [68]Animal studyEAE rats (MS mouse model)MaR1Intraperitoneal injectionSuppression of various pro-inflammatory cytokines, ↓ number of Th1 cells ↑ of Tregspolarization of macrophages towards an anti-inflammatory phenotype Prüss H et al., 2013 [69]Human studyMS patientsRvD1NDP1CSF levels↑ of RvD1Only detection of NDP1Kooij G et al., 2020 [70]Human studyNMOSD patientsRvD1LTB4CSF levelsRvD1 ↓LTB4 ↑Luo B et al., 2016 [73]Animal studyEAN (experimental autoimmune neuritis) modelRvD1Intraperitoneal injectionMacrophage phagocytosis of apoptotic T cells in PNS, ↑ TGFβ by macrophages, ↑ local Treg cell counts, and promotion of inflammation resolution and disease recoveryAnXA1: Annexin A1; AT-NPD1: aspirin-triggered NPD1; BCCAO: bilateral common carotid artery occlusion; DHA: docosahexaenoic acid; EAE: Experimental Autoimmune Encephalitis; EAN: Experimental Autoimmune Neuritis; FPR: formyl-peptide receptor; LXA4: Lipoxin A4; LXA4 ME: Lipoxin A4 Methyl Ester; LTB4: Leukotriene B4; MaR1: Maresin1; NPD1: Neuroprotectin D1; PNS: peripheral nervous system; RvD1: Resolvin D1.

### 3.3. Specialized Pro-Resolving Mediators in Neurodegenerative Diseases

AD is the most common type of dementia; a growing body of evidence suggests that inflammation is involved in its pathogenesis. Epidemiological studies suggest that the use of anti-inflammatory drugs is associated with a lower incidence of AD; however, clinical trials with anti-inflammatory drugs have not been successful [74]. Given these premises, the possibility of promoting resolution rather than inhibiting inflammation looks appealing. 

The potential benefit of working on inflammation resolution is supported by several observations. First of all, a shift in the LM profile in the CSF from pro-resolving to pro-inflammatory occurs as AD progresses: in a recent study, liquid chromatography–tandem mass spectrometry was used to analyze pro-resolving and pro-inflammatory LMs in the CSF of patients with cognitive impairment ranging from subjective impairment to a diagnosis of AD; LMs profile correlated to cognition, CSF tau, and β-amyloid. RvD4, RvD1, NPD1, MaR1, and RvE4 were lower in AD and/or mild cognitive impairment (MCI) compared to subjective cognitive impairment (SCI); on the other hand, pro-inflammatory mediators were higher in AD and MCI [75]. Similarly, it was found that the levels of the MaR1, NPD1 and RvD5, were lower in the entorhinal cortex of AD patients as compared to age-matched controls, while levels of the pro-inflammatory prostaglandin D2 (PGD2) were higher in AD [76]. In addition, RvD4 showed a negative correlation to AD tangle biomarkers and positive correlations to cognitive test scores [75]. Similar findings have been reported in mice, where SPMs in the brain cortex were substantially lower in mice with an APOE4 genotype [77]. The finding that SPMs receptors are increased in the AD brain in post-mortem studies and correlate to Braak stages, suggests a prominent role of resolution pathways; the increase in these receptors may either represent a primary factor in the pathogenesis of the disease or a consequence of failed resolution [78]. The same study group investigated age-related changes in the LM profile in the APP knock-in (APP KI) mouse model of AD, concluding that the brain lipidome appeared to be modified preferentially during aging as compared to amyloid pathology, as the oldest age group was the one with the greatest increase in LMs, despite an early onset of Aβ pathology [79]. In this case, the SPMs biosynthetic enzymes were found to be increased, while their receptor expression decreased in the aged App KI mice, in disagreement with their previous work [78] on AD patients. The discrepancy may be explained by the fact that the stage of AD pathology in 18-month-old App KI mice is likely less advanced compared to that seen in human post-mortem brains [79].

#### 3.3.1. SPMs Administration in AD Models

Several in vivo mouse studies support the potential benefit deriving from SPM use in AD. Table 3 provides a summary of in vivo studies on SPMs in neurodegenerative diseases. When a mixture of the SPMs including RvE1, RvD1, RvD2, MaR1 and NPD1 was administered to mice via intranasal delivery, an amelioration of memory deficits occurred, together with a restoration of gamma oscillation deficits, and a prominent decrease in microglial activation [80]. Intraperitoneal injection of RvE1 and LXA4, alone or in combination, increased the concentration of RvE1, LXA4, and RvD2 in the hippocampus of a murine model, reversed the inflammatory process and decreased the neuroinflammation associated with Aβ pathology; the levels of SPMs in the hippocampus of 5xFAD mice were in fact shown to be significantly lower than in wild-type mice [20]. Similarly, intracerebroventricular administration of LXA4 was able to inhibit the inflammatory response induced by β-amyloid in the cortex and hippocampus of experimental mice, in particular, the production of IL-1b and TNFa [81]. Other than LXA4, the effect of aspirin-triggered LXA4 (ATL) has been investigated too; ATL is generated after the acetylation of COX-2, and displays the same anti-inflammatory activity as the native lipoxins and is more resistant to metabolic inactivation [82]. Subcutaneous injection of ATL was able to reduce NF-kB activation and levels of proinflammatory cytokines and chemokines, as well as create an anti-inflammatory cerebral milieu, resulting in the recruitment of microglia in an alternative phenotype. Such microglia showed improved phagocytic function towards Aβ, ultimately leading to a reduction in synaptotoxicity and improvement in cognition [83]. Not only was ATL demonstrated to enhance the cognitive performance of 3xTg-AD mice and reduce Aβ load, but also to decrease the levels of phosphorylated-tau (p-tau) [84]. Furthermore, intracerebroventricular supply of MaR1 improved the cognitive decline of experimental mice; MaR1 was able to attenuate microglial activation, reduce pro-inflammatory cytokines in favor of anti-inflammatory ones, and up-regulate the levels of proteins related to survival pathways including PI3K/AKT, ERK and down-regulate the levels of proteins associated with inflammation, autophagy, and apoptosis pathways, such as p38, mTOR and caspase 3 [85]. NPD1 seems to play a role in decreasing inflammatory signaling in AD [86,87,88]; however, to our knowledge, NPD1 alone has never been administered to mice.

The potential role of AnxA1 in the AD murine model has been investigated too. AnxA1 is a pro-resolving mediator that helps to restore the integrity of the BBB and inhibit microglial activation in the brain; interestingly, these functions depend on AnxA1 integrity, and enzymatic cleavage generates pro-inflammatory fragments [89]. When AnxA1 level was measured in the blood and CSF of patients with AD and behavioral variant of frontotemporal dementia (bvFTD), reduced plasma levels of AnxA1 were observed in bvFTD compared to AD and controls, while no difference was shown in the CSF; moreover, a significant cleavage of AnxA1 in PBMCs in both dementia groups was shown [89]. A link between AnxA1, neuroinflammation and amyloid pathology is further suggested by the identification of elevated cleaved AnxA1 in the brains of patients with neurodegenerative dementias including AD, positively correlating with amyloidogenic brain Aβ, inflammatory and pro-apoptotic markers [90]. However, intact AnxA1 protein was found to be increased in the brain of both AD patients and animal models and induce the clearance and degradation of the amyloid-β peptide in vitro by acting on formyl peptide receptor-like 1 (FPRL1) [91]. The increases in ANXA1 observed in AD brains suggest that upregulation of AnxA1 could represent an adaptive response of microglia during inflammatory conditions and an attempt to turn down inflammation at the early disease stage; in later stages with chronic production of Aβ and pro-inflammatory cytokines, microglia change their neuroprotective phenotype in favor of a more pro-inflammatory activation state [91]. Still, surprisingly, when APP/PS1 double-transgenic AD mice were treated for 20 weeks with the anti-inflammatory FPR2 agonist Ac2-26, Ac2-26-treatment did not show any beneficial effect [92]. As previously mentioned, AnxA1 protects against BBB breakdown in AD: treatment with human recombinant ANXA1 (hrAnxA1) in the murine brain endothelial cell line bEnd.3 was able to rescue β-amyloid 1–42 -induced BBB disruption via inhibition of RhoA-ROCK signaling pathway [93]. Similarly, intravenous injection of hrAnxA1 was able to decrease BBB permeability, and reduce β-amyloid load and p-tau build-up in 5xFAD mice and Tau-P301L mice; in addition, the prolonged treatment with hrAnxA1 reduced the memory deficits and increased synaptic density in young 5xFAD mice [94].

Few studies have examined the neuroinflammation-modulating effects of *n*-3 PUFA feeding in the AD murine model; one study reported fish oil feeding managed to attenuate neuroinflammatory gene expression; however, no alteration in the levels of SPMs, brain eicosanoids or docosanoids was detected [95]. When investigating the ability of PUFAs to influence the production of SPMs in AD patients, a randomized, placebo-controlled trial found unchanged levels of the SPMs LXA4 and RvD1 in the group supplemented with *n*-3 FAs, whereas a decrease was documented in the placebo group, indicating that PUFA supplementation managed to prevent reduction in SPMs released from PBMCs [96]. Another work pointed to the ability of n3-PUFA to increase amyloid-β phagocytosis and RvD1 in patients with MCI [97].

#### 3.3.2. SPMs and Sphingosine Kinase

As a conclusive remark on AD and pro-resolving mediators, SPMs have been found to be regulated by sphingosine kinases (especially SphK1) that act by monitoring COX-2, a potent inhibitor of SPMs production [98]. SphK1 generates *N*-acetyl sphingosine (*N*-AS) from acetyl-CoA and sphingosine; *N*-AS then acetylates serine 565 (S565) of COX-2, and the *N*-AS-acetylated COX-2 induces the production of SPMs [99]. In a mouse model of AD, microglia showed a reduction in *N*-AS generation, leading to decreased acetyl-S565 COX2 and SPMs production; mouse treatment with *N*-AS increases acetylated COX-2 and *N*-AS-triggered SPMs in microglia, leading to resolution of neuroinflammation, an increase in microglial phagocytosis, and improved memory [99].

#### 3.3.3. SPMs in Neurodegenerative Disorders Other Than AD

While most literature on SPMs and neurodegenerative disorders focuses on AD, neuroinflammation is also one of the hallmarks of Parkinson’s disease (PD) and may play a role in midbrain dopamine (DA) neuron degeneration. Still, the effects of stimulating the resolution of inflammation in PD remain largely unexplored. Both in vitro [100,101] and in vivo models seem to point to a possible role of SPMs. In a lipopolysaccharide (LPS)-induced rat model of PD the effects of intrathecal injection of RvD2 on substantia nigra pars compacta (SNpc) were studied; RvD2 was shown to recover neural injury by suppressing inflammatory mediator expression [101]. In fact, LPS-induced inflammation in SNpc increased the expression of NO, iNOS, TNF-a, IL-1, IL-18, IL-6, IL-1b, ROS production, the translocation of NF-kB p65, IkBa, and IKKb expression in glial cells; after injection of RvD2, the treatment prevented development of behavioral defects and TLR4/NF-kB pathway activation. Another study on rats overexpressing human α-synuclein (Syn) demonstrated that prior to nigral degeneration they display altered DA neuron properties, and reduced striatal DA outflow and motor deficits; these early alterations are coupled with microglia activation and perturbations in inflammatory and pro-resolving mediators, namely IFN-γ and RvD1 [102]. When early and chronic intraperitoneal injection of RvD1 was provided, central and peripheral inflammation, as well as neuronal dysfunction and motor deficits were prevented. Interestingly, the same work demonstrated that endogenous RvD1 is decreased in human patients with early PD [102]. Supporting the role of SPMs in PD, homozygous missense variants in the AnxA1 were recently suggested to cause parkinsonism by leading to extracellular *SNCA* accumulation, neuroinflammation, as well as defects in intracellular signaling pathways and synaptic plasticity; however, such mutations seem to be exceedingly rare, and pathogenicity could not be further explored [103].

Other than AD and PD, a third relevant neurodegenerative pathology is amyotrophic lateral sclerosis (ALS). To our knowledge, no in vivo study on the role of SPMs in ALS has been performed. Although the cause of neuronal degeneration in ALS has not been fully elucidated, there is evidence of macrophage and T cell infiltration into the spinal cord, which may be responsible for motor neuron death. In ALS macrophages, aggregated superoxide dismutase-1 (SOD-1) stimulated the expression of inflammatory cytokines, including IL-1β, IL-6, and TNF-α; it was shown that RvD1 was able to inhibit macrophage IL-6 and TNF-α production, thus suppressing inflammation [104]. Another study investigated the effects of MaR1 on motor neuron cell death, finding it protected motor neuron-like NSC-34 cells against serum-free and SOD1^G93A^ or TDP-43^A315T^-induced cell death, as well as H_2_O_2_- or tunicamycin-induced cell death [105]. molecules-27-04836-t003_Table 3Table 3Summary of in vivo studies on SPMs in neurodegenerative diseases.ReferenceType of StudyModelPro-Resolving MediatorDelivery (Or Measurement If the Study Was Non-Interventional)OutcomeDo K V et al., 2022 [75]Human, non-interventional Patients with AD, MCI, SCIRvD4CSF levels of RvD4Negative correlation to AD tangle biomarkers, and positive correlations to cognitive test scoresZhu M. et al., 2016 [76]Human studyPatients with ADMaR1, NPD1, RvD5Postmortem tissue samples from the entorhinal cortex↓ concentration of pro-resolving mediators in the entorhinal cortex of AD patients as compared to age-matched controls, while levels of the pro-inflammatory prostaglandin D2 were higher in ADMartinsen A. et al., 2019 [77]Animal studyAPOE4 Female miceVarious SPMsBrain postmortem tissue samples ↓ SPMs in mice with the APOE4 genotypeEmre C. et al., 2020 [78]Human studyPatients with AD SPMs receptorsBrain postmortem tissue samples↑ SPMs receptorsEmre C, Do K V. et al., 2021 [79]Animal studyAPP KI mouse model of ADLMs profileBrain postmortem tissue samples↑ microglia proliferation starting from a young age in the App KI mice, while ↓ astrocyte numbers in older agesBrain lipidome appears to be modified preferentially during aging as compared to amyloid pathology, as the oldest age group was the one with the greatest increase in LMs, despite an early onset of Aβ pathologyEmre C, Arroyo-García et al., 2022 [80]Animal studyMurine model of ADRvE1, RvD1, RvD2, MaR1 and NPD1IntranasalAmelioration of memory deficits; restoration of Gamma oscillation deficits; ↓ microglial activationKantarci A. et al., 2017 [20]Animal studyMurine model of ADRvE1 and LXA4Intraperitoneal↑ RvE1, LXA4, and RvD2 in the hippocampus; reversing of the inflammatory process, ↓ neuroinflammation Wu J. et al., 2011 [81]Animal studyMurine model of ADLXA4IntracerebroventricularInhibiting the inflammatory response induced by β-amyloid in the cortex and hippocampus (in particular, production of IL-1b and TNFa)Serhan CN., 2005 [82]Animal studyMurine model of ADATLSubcutaneous↓ NF-kB activation and levels of proinflammatory cytokines and chemokines; creating an anti-inflammatory cerebral milieu, resulting in the recruitment of microglia in an alternative phenotypeMedeiros R. et al., 2013 [83]Animal studyMurine model of ADATLSubcutaneous↓ phosphorylated-tau (p-tau)Yin P. et al., 2019 [85]Animal studyMurine model of ADMaR1IntracerebroventricularImproving cognitive decline of experimental mice: attenuating microglial activation, ↓ the pro-inflammatory cytokines in favor of anti-inflammatory ones, and ↑ the levels of proteins related to survival pathway including PI3K/AKT, ERK; ↓ levels of proteins associated with inflammation, autophagy, and apoptosis pathways, such as p38, mTOR and caspase 3Schröder N et al., 2020 [92]Animal studyMurine model of AD Ac2-26 Intraperitoneal injectionNo beneficial effectPark JC et al., 2017 [93]In vitro and in vivo (animal study)Aβ-42 treated murine brain endothelial cell line bEnd.3;Murine model of AD Human recombinant ANXA1; ANXA1Administration of human recombinant ANXA1 in Aβ-42 treated murine brain endothelial cell line bEnd.3; *ANXA1 levels on blood of murine model of AD*rescuing β-amyloid 1–42 -induced BBB disruption via inhibition of RhoA-ROCK signaling pathway in brain endothelial cell line bEnd.3;↓ ANXA1 in a murine model of ADRies M. et al., 2021 [94]Animal studyMurine model of ADHuman recombinant AnxA1 Intravenous injection↓ β-amyloid load and p-tau build-up in 5xFAD mice and Tau-P301L mice; prolonged treatment reduced the memory deficits and increased synaptic density in young 5xFAD miceTian Y. et al., 2015 [101]Animal studyRat model of PDRvD2Intrathecal injection on substantia nigra pars compactarecovering neural injury by suppressing inflammatory mediator expressionKrashia P. et al., 2019 [102]Animal studyRats overexpressing human α-synuclein (Syn)RvD1Chronic intraperitoneal injectionpreventing central and peripheral inflammation, as well as neuronal dysfunction and motor deficits AD: Alzheimer Disease; Ac2-26: annexin/lipocortin 1-mimetic peptide; ATL: Aspirin-triggered LXA4; LXA4: Lipoxin A4; LMs: Lipid Mediators; MaR1: Maresin1; MCI: Mild cognitive impairment; *N*-AS: *N*-acetyl sphingosine; n3-PUFAs: omega-3 polyunsaturated fatty acids; NPD1: Neuroprotectin D1; PD: Parkinson Disease; RvD1: Resolvin D1; RvD4: Resolvin D4; RvD5: Resolvin D5; RvE1: Resolvin E1; SCI: Subjective cognitive impairment.

## 4. Materials and Methods

The PubMed library was searched for journal articles published in English up to 4 May 2022; we used the entry words “pro-resolving mediators”, “resolvins”, “maresins”, “annexins”, “lipoxins”, “protectins”, “neuroinflammation”, “central nervous system”, “glial cells”, and “Alzheimer”, “Parkinson”, “neurodegenerative”, “stroke”, “cerebrovascular”. We purposefully focused on findings in human studies and on pre-clinical studies which have been implemented in the animal model, as pre-clinical animal studies more closely resemble possible future applications in clinical practice. In vitro studies have not been discussed in this work. However, a concise overview of the main in vitro models currently used in the field of neurological diseases and SPMs research has been provided in Table 4.

## 5. Conclusions

Inflammation is a reaction to a harmful agent, physiologically self-contained thanks to the intervention of endogenous molecules which promote its resolution. If persistent or dysregulated, inflammation itself becomes noxious for human tissues.

Chronic, low-grade inflammation of the CNS is considered the pathophysiological foundation of many neurological disorders and of the neurodegenerative processes themselves.

In this picture, pro-resolution is a spontaneous collateral biochemical mechanism led by SPMs in the inflamed tissues and the identification of these molecules contributed to the understanding of the inflammation processes; their use in the clinical setting could potentially be an important tool for clinicians. 

We have analyzed the bulk of the evidence in this article on the role of SPMs in the control of inflammation processes in several models of the most important neurological disorders. This amount of evidence has moved an interest among patients and physicians for the clinical use of SPMs. 

Unfortunately, at the moment, this interest cannot be certainly defined as evidence-based.

A preliminary problem to be explored for future clinical studies relates to the route of administration of SPMs. In fact, regarding the administration of SPMs, there is at present no evidence on whether they can actually cross the BBB, although their characteristic of small lipophilic molecules makes this possibility plausible, similarly to what is known for their precursors DHA and EPA [74].

A second aspect concerns the definition of precise and clinically meaningful outcome measures, specific for the diseases to be treated.

Finally, little is known about the toxicology of SPMs, although no side effects have been reported so far. However, since research on SPMs is relatively new, an effort should be made to conduct future studies on safety, in order to rule out possible harmful effects. This is even more important if we consider that a few, not recent, studies on ω-3 PUFAs, which are SPMs precursors, have pointed out that they may impact platelet aggregation and reduce the immune response to infections [106,107,108,109].

Having in mind these considerations, we believe that the information coming from animal studies should prompt investigators and industry to fill the scientific gap with robust clinical studies on SPMs, which are tremendously needed.

It is likely that the clinical use of SPMs will not be as potent as that of anti-inflammatory drugs, but their action is likely more physiological, and it could probably be better tolerated by patients. In addition, their effects could be potentiated by the synergic action of other “natural” approaches to the control of chronic low-grade inflammation, such as those based on nutrition and lifestyle.

## Figures and Tables

**Table 4 molecules-27-04836-t004:** Overview of the main in vitro models currently used in the field of neurological disease and SPMs research.

In Vitro Model	Brief Model Description
** *Cerebrovascular diseases* **	
OGD of rat cortical neurons	Primary cortical neurons are subjected to OGD mimicking ischemic injury.
OGD/R of BV2 murine microglial cell	BV2 murine microglial cells are subjected to OGD mimicking ischemic injury, with subsequent reoxygenation and exposure to a glucose-containing medium.
OGD/R of rat astrocytes	Primary astrocytes are exposed to OGD mimicking ischemic injury, with subsequent reoxygenation and exposure to a glucose-containing medium.
** *Immune-mediated demyelinating disorders* **	
Imiquimod and ssRNA40- stimulated PBMCs	Freshly isolated PBMCs are stimulated with Imiquimod (TLR 7 agonist) and ssRNA40 (TLR 8 agonist) to induce inflammatory changes.
Mixed glial cell model	This model is meant to study the expression of inflammatory mediators and myelin genes under inflammation; mixed glial cell cultures are treated with a combination of pro-inflammatory cytokines to create an inflammatory environment.
Co-culture studies	Mouse brain microglial cells are co-cultured with rat oligodendrocyte progenitor cells and then processed for expression of myelin genes. Co-culture systems allow studying the interactions between cell populations.
** *Neurodegenerative diseases* **	
**AD models**	
AD patients PBMCs	PBMCs drawn from the venous blood of AD patients.
Aβ40- or Aβ42-exposed PBMCs	PBMCs are isolated from patients’ peripheral venous blood and incubated with Aβ40 or Aβ42 to mimic the AD environment.
Aβ42-exposed human CHME3 microglial cells	Human microglial cell line CHME3 has also been employed and incubated with Aβ42.
Aβ42-treated HNG	Co-cultures of neurons derived from mice and mouse brain mixed glial cells, subsequently stimulated by Aβ42. Co-culture of human cells has been employed too (primary human neuronal-glial co-culture).
Aβ-stimulated BV2 microglial cells	Mouse microglial cell line BV2 gets incubated with Aβ42.
HNG transfected with βAPPsw	HNG cells may either be challenged with Aβ42 oligomeric peptide as described above or transfected with beta amyloid precursor protein (βAPP)sw to mimic AD in vitro.
STS-induced apoptosis in neuroblastoma cells	Neuroblastoma cell line SH-SY5Y represents a model of human neuronlike cells. To study neuronal survival, apoptosis can be induced by incubating the differentiated SH-SY5Y cells with STS.
**PD models**	
LPS-induced murine microglial cells	Rat microglial cells incubated with the addition of LPS, which induces inflammatory changes.
MPP+ -treated PC12 pheochromocytoma cells	MPTP is an environmental toxin that specifically damages DA neurons; the same applies to its metabolite MPP+, explaining why they are commonly used to obtain in vitro PD models. In this case, PC12 rat pheochromocytoma cells are treated with MPP+.
**ALS models**	
Fibrillar wild type SOD-1-stimulated PBMCs	In ALS PBMCs, in vitro aggregated SOD-1 is used to stimulate the expression of inflammatory cytokines.
SOD1G93A or TDP-43A315T- transfected motor neuron-like NSC-34	NCS-34 cells are transfected with SOD1G93A plasmid or TDP-43A315T plasmid, which induces cell death thus providing a model of motor neuron degeneration.

Aβ40: Amyloid Beta 1–40; Aβ42: Amyloid Beta 1–42; AD: Alzheimer’s Disease; ALS: Amyotrophic Lateral Sclerosis; (βAPP)_sw_: Swedish double mutation APP695_sw_, K595N-M596L; DA: dopamine; HNG: Human Neuronal-Glial co-culture; LPS: Lipopolysaccharide; MPP+: 1-methyl-4-phenylpyridium; MPTP: 1-methyl-4-phenyl-1,2,3,6-tetrahydropyridine; NSC-34: Neuroblastoma spinal cord 34; OGD: oxygen-glucose deprivation; OGD/R: oxygen-glucose deprivation/reoxygenation; PBMCs: Peripheral Blood Mononuclear Cells; PD: Parkinson’s disease; SOD-1: Superoxide Dismutase-1; STS: Staurosporine; TDP-43: TAR DNA-binding protein 43; TLR: Toll-like receptor.

## Data Availability

Not applicable.

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
