# Peer review of "Specialized Pro-Resolving Mediators in Neuroinflammation: Overview of Studies and Perspectives of Clinical Applications"

_molecules, 2022, doi:10.3390/molecules27154836_

Round 1

Reviewer 1 Report

Dear Valente et al.,

In this article, the authors provide a comprehensive and in-depth review of pro-resolving mediators in neuroinflammation. The details of the synthesis, mechanisms of action and the role of various poly-unsaturated fatty acids (PUFAs)-derived mediators in diseases with neuroinflammation components is expansive and appropriate. While overall review is well-written, some minor changes regarding references are advised.

It is recommended that, whenever possible, writers give credit to the scientists who performed original research, rather than citing review articles. For example, on the second page of line 49, authors write "Aspirin can trigger their 49 biosynthesis thanks to its capacity to promote the formation of lipid mediators via lipo- 50 oxygenation [1]. While the authors cite the review article by Fredman G and Serhan CN, the original experiment was performed by Clària J and Serhan CN 1995 (Clària J, Serhan CN. Aspirin triggers previously undescribed bioactive eicosanoids by human endothelial cell-leukocyte interactions. Proc Natl Acad Sci U S A. 1995 Oct 10;92(21):9475-9. doi: 10.1073/pnas.92.21.9475. PMID: 7568157; PMCID: PMC40824.).

There are other instances like this: third page line 116 regarding LGR6 receptor should cite (Chiang N, Libreros S, Norris PC, de la Rosa X, Serhan CN. Maresin 1 activates LGR6 receptor promoting phagocyte immunoresolvent functions. J Clin Invest. 2019: doi: 10.1172/jci129448); line 117 regarding NPD1/PD1 should cite (Bang S, Xie YK, Zhang ZJ, Wang Z, Xu ZZ, Ji RR. GPR37 regulates macrophage phagocytosis and resolution of inflammatory pain. J Clin Invest. 2018;128(8):3568–82.), and line 118 regarding RvD5n-3 DPA should cite (Flak MB, Koenis DS, Sobrino A, Smith J, Pistorius K, Palmas F, et al. GPR101 mediates the pro-resolving actions of RvD5n-3 DPA in arthritis and infections. J Clin Invest.).

Similarly, 4th page lines 139-142 regarding neuroinflammation and neurodegenerative diseases should cite each original articles that demonstrate the said findings.

Page 3 line 110, citation [2] only cites citation 5, hence unnecessary.

While it seems minor, we should try to give scientists who performed the experiments credit, rather than review articles.

Other minor advised changes are:

Are Chem23 synonymous to ChemR23? In lines 102 and 105 it is written as Chem23, while in line 181 it is written as ChemR23.

Line 195, instead of “yet not”, “not yet” makes more sense.

If the aforementioned recommendations are addressed, the review seems to be a significant addition to the field.

Best wishes,

Joon Seo

Author Response

Dear Joon Seo,

Thank you for your words of appreciation and for your on-point comments. We fixed the references as per your suggestion. We confirm “Chem23” being a typo, we were indeed referring to ChemR23; the text has been updated with the correction. We performed a further English revision, and mistakes like the one you pointed out in line 195 have been taken care of. We do hope that your suggestions have been carefully met.

Best regards,

Professor Valente et al.

Reviewer 2 Report

Brief summary

The manuscript entitled "Specialized pro-resolving mediators in neuroinflammation: overview of studies and perspectives of clinical applications" gives a broad and sufficiently detailed review of the field of interest, namely the role of specialized pro-resolving mediators (SPMs) in neuroinflammation and major neurological diseases. The review is comprehensive, reads well, and covers the most significant achievements and remaining open questions of this intensively emerging field.

Broad comments

Areas of strength

The manuscript comprehensively describes the various SPMs regarding their metabolism, respective receptors, and pathways. It summarizes in vivo studies on SPMs in ischemic cerebrovascular events, stroke, neurodegenerative diseases, and neurological immune-mediated disorders. It also provides well-structured useful tables on these numerous studies.

Areas of weakness

The manuscript does not cover any in vitro studies; however, the review would probably benefit from a brief table with the most relevant in vitro model systems.

Specific comments 

1. Please give the resolution of abbreviations at their first site of mention, for example, PMs at line 39 and AD at line 561.

2. Please correct the missing R in ChemR23 at lines 102 and 105.

3. The G-protein-coupled seven-transmembrane receptors are usually abbreviated as GPCR and not as 7-TMs, see line 97 and 106.

4. Please check for typos such as "od" at line 624.

Author Response

Dear referee,

Thank you for your words of appreciation and for your on-point comments. Following your suggestions, we added a table on the main in vitro models adopted in the research field of neurological disorders and SPMs. Far from being comprehensive, we do hope it provides a simple but useful outline of the most relevant in vitro systems. Formal issues, including typos and appropriate resolution of abbreviation, have also been taken care of. We do hope that your suggestions have been carefully met.

Best regards,

Professor Valente et al.